

# Common metabolic constraints on dive duration in endothermic and ectothermic vertebrates

April Hayward[1], Mariela Pajuelo[1], Catherine G. Haase[2], David M. Anderson[1] and James F. Gillooly[1]

[1] Department of Biology, University of Florida, Gainesville, FL, USA
[2] School of Natural Resources and Environment, University of Florida, Gainesville, FL, USA

## ABSTRACT

Dive duration in air-breathing vertebrates is thought to be constrained by the volume of oxygen stored in the body and the rate at which it is consumed (i.e., "oxygen store/usage hypothesis"). The body mass-dependence of dive duration among endothermic vertebrates is largely supportive of this model, but previous analyses of ectothermic vertebrates show no such body mass-dependence. Here we show that dive duration in both endotherms and ectotherms largely support the oxygen store/usage hypothesis after accounting for the well-established effects of temperature on oxygen consumption rates. Analyses of the body mass and temperature dependence of dive duration in 181 species of endothermic vertebrates and 29 species of ectothermic vertebrates show that dive duration increases as a power law with body mass, and decreases exponentially with increasing temperature. Thus, in the case of ectothermic vertebrates, changes in environmental temperature will likely impact the foraging ecology of divers.

## INTRODUCTION

The length of time that air-breathing vertebrate divers can remain submerged is an important constraint on their foraging activities, and perhaps ultimately on their fitness (*Butler & Jones, 1982*; *Kooyman, 1989*; *Stephens et al., 2008*; *Andrews & Enstipp, in press*). Consequently, vertebrates display a wide variety of behavioral, morphological, and physiological adaptations to enhance dive capacity. For example, endothermic vertebrates that dive show a relatively high capacity to store oxygen, use anaerobic metabolism, and reduce oxygen demand in non-vital tissues to prolong dive duration (*Boyd, 1997*; *Butler & Jones, 1997*; *Kooyman & Ponganis, 1998*). With respect to oxygen storage, cetaceans have higher densities of myoglobin in their tissues of a molecular form that has evolved to be more stable at depth (*Mirceta et al., 2013*). Despite such specialization, maximum dive duration in endotherms varies somewhat predictably with body mass based on the amount of oxygen stored in the body and the rate at which oxygen is consumed (*Butler & Jones, 1982*; *Schreer & Kovacs, 1997*; *Halsey, Blackburn & Butler, 2006*; *Halsey, Butler & Blackburn, 2006*; *Stephens et al., 2008*). However, the extent to which dive duration

Corresponding author
James F. Gillooly, gillooly@ufl.edu

may similarly vary with body mass in ectotherms as predicted by the oxygen store/usage hypothesis is less clear. Recent studies concluded that ectotherms fail to conform to expectations, and attributed this to ectotherms possessing characteristics that weaken any allometric constraints associated with diving (e.g., the ability to use aquatic respiration to supplement oxygen stores, and to enter thermally-induced hypo-metabolic states, to extend dives) (*Brischoux et al., 2008*; *Campbell et al., 2010*). In the case of aquatic respiration, some air-breathers may respire in water using lungs, whereas others may respire through their integument (i.e., "cutaneous respiration"; *Lenfant & Johansen, 1972*; *Feder & Burggren, 1985*). Yet, the potentially confounding effects of temperature on oxygen consumption rates and thus dive duration have not yet been examined in vertebrate ectotherms.

Here we present broad-scale analyses of dive duration in both endothermic and ectothermic vertebrates. We compare the predicted body mass and temperature dependence of dive duration based on the oxygen store/usage hypothesis to analyses of extensive empirical data compiled from the literature. The model and results presented here build on previous work in endotherms (*Butler & Jones, 1982*; *Kooyman, 1989*; *Stephens et al., 2008*) to examine the relationship in ectotherms, particularly with respect to temperature.

We begin by examining the predicted body mass and/or temperature dependence of dive duration based on the oxygen store/usage hypothesis. This hypothesis stipulates that dive duration ($t_D$) is a function of the total amount of oxygen carried by an organism on its dive ($TO_2$) divided by the rate at which that oxygen is used (i.e., its metabolic rate, $B$) (*Butler & Jones, 1982*; *Kooyman, 1989*). Thus,

$$t_D = TO_2/B \tag{1}$$

where oxygen storage capacity scales approximately linearly with body mass ($M$) and is independent of temperature (*Lindstedt & Calder III, 1981*; *Stephens et al., 2008*; *Campbell et al., 2010*) such that:

$$TO_2 = aM^1, \tag{2}$$

where $a$ is a constant that describes the amount of oxygen that can be stored per gram of body mass (*Lindstedt & Calder III, 1981*; *Campbell et al., 2010*). Oxygen consumption rate in Eq. (1), however, has both a body mass and temperature dependence such that:

$$B = c \cdot M^d \cdot e^{0.12Tc}, \tag{3}$$

where $c$ is a constant describing the amount of oxygen consumed per unit body mass, $d$ describes the scaling of whole organism oxygen consumption rate with body mass ($M$), and $e^{0.12Tc}$ describes the exponential temperature dependence of oxygen consumption rate that is roughly equivalent to a $Q_{10}$ of 2.5 (*Charnov & Gillooly, 2003*). Thus, we expect that the oxygen store/usage hypothesis should have both a body mass and temperature dependence such that:

$$t_D = a \cdot c^{-1} \cdot M^{b-d} \cdot e^{-0.12Tc}. \tag{4}$$

Note that the body mass dependence of oxygen consumption rate may differ between endotherms and ectotherms (*Gilloolly et al., 2016*), and thus the body mass dependence of dive duration may also differ. Still, the scaling exponent for whole organism oxygen consumption rate shown in Eq. (4) (i.e., "*d*") is expected to be close to a value of 3/4 (*Isaac & Carbone, 2010*) and thus considerably less than the scaling exponent for oxygen storage (i.e., "*b*") of roughly 1. The effect of temperature on oxygen consumption rate, though, appears to be more similar in both groups (*Gilloolly et al., 2001*). We evaluate the body mass and temperature dependence of dive duration from Eq. (4) for a diverse assortment of mammals, birds, reptiles, and amphibians (Appendix S1).

## METHODS

### Data collection

Data on median and maximum dive duration, body mass, and temperature were obtained from previously published studies for 181 species of endotherms and 29 species of ectotherms (Appendix S1). Leatherback turtles (*Dermochelys coriacea*) were excluded from consideration since they are functionally endothermic (*Penick et al., 1998*; *Southwood et al., 2005*; *Bostrom & Jones, 2007*), and direct measures of body temperature were not available. The body temperature of this species may be as much as 18 °C higher than ambient temperatures (*Frair, Ackman & Mrosovsky, 1972*).

For ectotherms, we used direct measures of body temperature when available, but otherwise we used ambient environmental temperatures as a proxy for body temperatures. For endotherms, we used species-specific measures of body temperatures for mammals and birds when available, or the mean body temperature of species from the same genus (*Clarke & Rothery, 2008*). If both of these estimates were not available, we used the mean body temperatures of birds (41.5 °C) and mammals (37 °C) (*Clarke & Rothery, 2008*) (Appendix S1). For body mass, we used the values of study subjects, or if unavailable, estimates of adult body masses from other sources (Appendix S1).

### Analyses

We evaluated the body mass and temperature dependence of median and maximum dive duration using Bayesian generalized linear mixed-models (*Lunn et al., 2000*; *Bolker et al., 2009*) implemented in R package *MCMCglmm* (*Hadfield, 2010*). In performing these analyses, we accounted for any non-independence due to shared evolutionary history by including a vertebrate supertree recently constructed by *Gilloolly et al. (2016)*, and by treating species as a random effect. This approach also accounted for effects of species sample sizes. To evaluate statistical models, we calculated conditional $R^2$ values for each model (*Nakagawa & Schielzeth, 2013*), and assessed model assumptions using diagnostic tests associated with phylogenetic analyses (*Paradis, Claude & Strimmer, 2004*; *Kembel et al., 2010*).

To best assess the temperature-dependence of median and maximum dive duration, we first performed analyses on the entire dataset. This dataset includes individuals of the same species measured at different temperatures. To best assess the mass-dependence of dive duration, we performed an additional set of analyses that was largely restricted to one point per species except when body mass differed substantially between males or females

**Table 1   Body mass and temperature dependence of dive duration in vertebrates.** Outputs from generalized linear mixed-models relating dive duration (median or maximum, in minutes) to body mass (natural log-transformed, in grams) and temperature (°C). Conditional $R^2$ values are reported, along with the slopes and intercepts of the relationships, and the sample sizes ($N$; with number of species). Results shown here represent the full dataset with multiple individuals for many species.

| Group | Resp. Var. | Intercept (95% CI) | ln(Mass) (95% CI) | Temp. (95% CI) | $R^2$ | $N$ (species) |
|---|---|---|---|---|---|---|
| Ectotherms | Med. | 4.00 (1.78,5.83) | 0.19 (0.06,0.32) | −0.13 (−0.16,−0.09) | 0.72 | 267 (29) |
| | Max. | 5.88 (3.42,8.21) | 0.09 (−0.09,0.26) | −0.11 (−0.13,−0.09) | 0.81 | 267 (29) |
| Endotherms | Med. | −1.65 (−7.51,3.79) | 0.42 (0.35,0.49) | −0.06 (−0.20,0.08) | 0.72 | 738 (181) |
| | Max. | 0.52 (−4.71,6.37) | 0.34 (0.30,0.42) | −0.07 (−0.21,0.07) | 0.62 | 738 (181) |
| All | Med. | 2.34 (0.94,3.82) | 0.37 (0.31,0.42) | −0.14 (−0.16,−0.11) | 0.38 | 1,005 (210) |
| | Max. | 3.34 (1.98,4.56) | 0.32 (0.27,0.38) | −0.12 (−0.14,−0.10) | 0.28 | 1,005 (210) |

of a species, or between juveniles and adults. For this second set of analyses, median and maximum dive durations were normalized to 30 °C for all species by assuming a $Q_{10}$ of 2.5 for oxygen consumption rate (*Gillooly et al., 2001*; *White, Phillips & Seymour, 2006*).

## RESULTS AND DISCUSSION

Results show support for the body mass and temperature dependence of dive duration predicted by the oxygen store/usage hypothesis (*Butler & Jones, 1982*; *Kooyman, 1989*; Eq. (4)). With respect to body mass, analysis of the full dataset showed that median and maximum dive duration increased as a power law with body mass in both ectothermic and endothermic vertebrates (Table 1). While scaling exponents varied from 0.09–0.42 depending on the group in question, for ectotherms the 95% confidence intervals of these scaling exponents included the 0.25 one would expect for dive duration (Table 1; Eq. (4)) based on the linear scaling of oxygen storage and a 3/4 power scaling of oxygen consumption rate (*Isaac & Carbone, 2010*; *Gillooly et al., 2016*; but see *White, Phillips & Seymour, 2006*). In the case of endotherms, however, the confidence limits of these analyses did not include 3/4 (but did include 2/3; Table 1). Still, when the dataset was restricted largely to one point per species, and adjusted to a common temperature, the scaling exponents for median and maximum dive duration ranged from only 0.21 to 0.34, in closer agreement with expectations from the oxygen store/usage hypothesis and the 3/4 scaling of oxygen consumption (Table 2; Eq. (4)). With respect to ectotherms, then, our results differ from previous studies showing no body mass dependence of dive duration (*Brischoux et al., 2008*; *Campbell et al., 2010*). We speculate that this is because we explicitly accounted for the effects of temperature.

A comparison of the body mass-dependence of dive duration in endothermic and ectothermic vertebrates shows both similarities and differences-both of which are qualitatively consistent with the oxygen store/usage hypothesis. First, body mass explained between 63–68% in temperature-adjusted median and maximum dive duration in both groups. However, the scaling exponents for these relationships in ectotherms (0.21, 0.22) were slightly lower than those of endotherms (0.31, 0.34; Fig. 1 and Table 2). This observation is consistent with work showing that the body mass scaling of oxygen consumption rate is steeper in ectotherms (exponents: 0.84–0.90) than in endotherms (0.70–0.74)

**Table 2  Body mass dependence of temperature-adjusted dive duration in vertebrates.** Outputs from generalized linear mixed-models relating temperature-adjusted dive duration (median or maximum) to body mass (ln-transformed, in grams) for ectothermic (amphibians and reptiles) and endothermic (mammals and birds) vertebrates. Conditional $R^2$ values are reported, along with the slopes and intercepts of the relationships, and sample sizes ($N$; with number of species).

| Group | Res. Var | Intercept (95%) | ln(Mass) (95% CI) | $R^2$ | $N$ (species) |
|---|---|---|---|---|---|
| Ectotherms | Med. | 0.12 (−1.88,2.34) | 0.22 (0.02,0.42) | 0.68 | 28 (28) |
| | Max. | 2.05 (−2.26,6.36) | 0.21 (−0.08,0.55) | 0.67 | 28 (28) |
| Endotherms | Med. | −2.49 (−3.94,−0.90) | 0.34 (0.23,0.42) | 0.63 | 187 (165) |
| | Max. | −1.02 (−2.51,0.42) | 0.31 (0.24,0.39) | 0.68 | 187 (165) |
| All | Med. | −1.47 (−2.72,−0.26) | 0.30 (−0.21,0.41) | 0.33 | 215 (193) |
| | Max. | 0.02 (−1.30,1.57) | 0.31 (0.22,0.42) | 0.22 | 215 (193) |

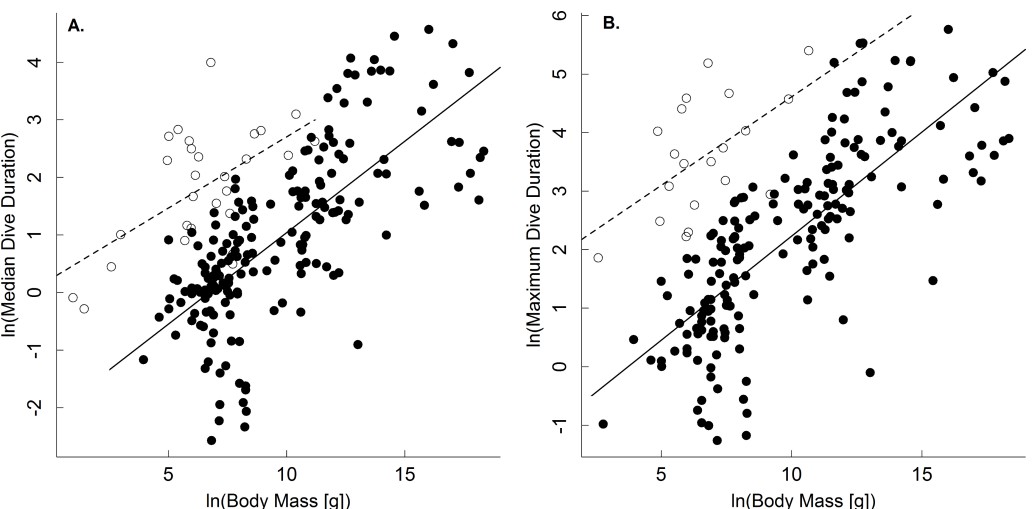

**Figure 1  The body mass dependence of dive duration in vertebrates.** The natural logarithm of median (A) and maximum dive duration (min.; B) as a function of the natural logarithm of body mass (g) for air-breathing endothermic (birds and mammals; closed points, solid line) and ectothermic vertebrates (reptiles and amphibians; open points, dashed line). Data were normalized to 30 °C assuming a $Q_{10}$ of 2.5. Most points represent a single species (see 'Methods,' and Table 2 for statistics).

(*Gillooly et al., 2016*). Similarly, the relatively high intercepts of the dive duration-body mass relationships in ectotherms as compared to endotherms are consistent with well-established differences in the oxygen consumption rates of ectotherms and endotherms. On average, whole organism oxygen consumption rates of endothermic vertebrates are approximately 1–2 orders of magnitude higher than ectotherms after accounting for any effects of temperature (*Gillooly et al., 2001*; *Brown et al., 2004*). As such, endotherms would be expected to use their oxygen stores more quickly than ectotherms, and have dive durations roughly an order of magnitude lower than ectotherms-as was observed (Fig. 2).

Dive duration was also shown here to vary systematically with temperature (Table 1 and Fig. 2). Note that while temperature has been shown to affect the dive duration of individual species (*Priest & Franklin, 2002*; *Storch et al., 2005*), no broad-scale analyses of temperature on dive duration have been previously undertaken. The temperature dependence of the

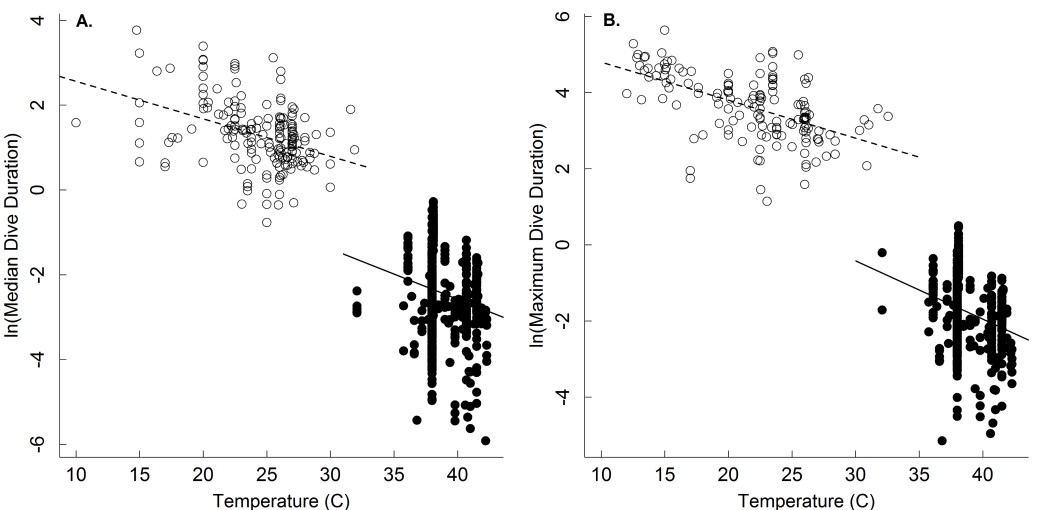

**Figure 2** **The temperature dependence of dive duration in vertebrates.** The natural logarithm of body mass-corrected median (A) and maximum (B) dive duration as a function of temperature (°C) for endothermic (birds and mammals; closed points) and ectothermic vertebrates (reptiles and amphibians; open points, dashed line). Many species are represented by multiple points, as described in the methods (see Table 1 for statistics).

oxygen store/usage hypothesis described by Eq. (4) has not been fully appreciated since the hypothesis has largely been applied to endotherms. Our results show that, at least in ectotherms, median and maximum dive duration decreases with increasing temperature as $e^{-0.11}$ and $e^{-0.13}$, respectively (Table 1 and Fig. 2)-similar to that described by Eq. (4). This equates roughly to a $Q_{10}$ of 2.5 such that dive duration will decrease by roughly 2.5-fold for every 10 °C increase in temperature. Thus, increases in environmental temperature could substantially reduce foraging time for ectothermic divers, and thus potentially affect individual fitness and population viability. The effect of any such increase would be most acute at warmer temperatures, where a smaller increase in temperature could have a greater effect on dive duration (see *Dillon, Wang & Huey, 2010*).

Together, these results point to the utility of the oxygen store/usage hypothesis for explaining some similarities and differences in dive duration among diverse vertebrates. Still, a deliberately simplified model such as this is perhaps most useful as a point of departure for examining species-specific adaptations for diving-both physiological and behavioral. Accounting for the effects of body mass and temperature on dive duration should be helpful in evaluating the benefit of such adaptations. More broadly, the model and results presented here demonstrate how considering the physiological effects of body size and temperature can reveal important insights into behavioral ecology (*Hayward, Gillooly & Kodric-Brown, 2012*).

## ACKNOWLEDGEMENTS

We thank F Brischoux and A Hein for helpful discussions in the early stages of this project. We also thank JP Gomez and E Mavrodiev for their assistance with statistical matters and phylogenetics, and A Clarke for graciously allowing us to use his dataset.

### Funding

The University of Florida provided support for Dr. April Hayward. No grants were used to support this work. The funders had no role in study design, data collection and analysis, decision to publish, or preparation of the manuscript.

### Competing Interests

The authors declare there are no competing interests.

### Author Contributions

- April Hayward and James F. Gillooly conceived and designed the experiments, performed the experiments, analyzed the data, wrote the paper, prepared figures and/or tables, reviewed drafts of the paper.
- Mariela Pajuelo performed the experiments, analyzed the data.
- Catherine G. Haase analyzed the data, prepared figures and/or tables, reviewed drafts of the paper.
- David M. Anderson performed the experiments, analyzed the data, prepared figures and/or tables.

### Data Availability

The raw data has been supplied as a Supplementary File.

### Supplemental Information

Supplemental information for this article can be found online at http://dx.doi.org/10.7717/peerj.2569#supplemental-information.

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
