# Peer review of "Common metabolic constraints on dive duration in endothermic and ectothermic vertebrates"

_PeerJ, doi:10.7717/peerj.2569_

## Round 0.1 · original submission · Minor Revisions

Three reviewers have assessed your manuscript and all agree that you have developed a very interesting dive model. Each of the 3 reviewers have made minor comments and I feel each of these comments would make your manuscript even better, so please in your rebuttal follow these comments. Once this has been done, your manuscript will be accepted.

Reviewer 1 ·

Basic reporting

The manuscript is well written and appears to conform to the Peer J requirements.

Experimental design

Appears to be rigorous.

Validity of the findings

The findings are robust but not surprizing.

Additional comments

In this study the authors compile a large dataset detailing dive durations, dive times, animal mass and temperature of ectothermic and endothermic vertebrates to investigate the allometric relationships between body mass or temperature and dive duration. The results confirm that dive duration increases with increasing body mass in both groups. However, there was no clear correlation with dive duration and temperature in endotherms but there was in ectotherms. I have no major concerns.

Concerns.
Pointing out that oxygen storage and body mass correlate isometrically while body mass and oxygen consumption correlate in a negative allometric way may be useful to add to the introduction.
Berenbrink and Campbell have shown that myoglobin is modified in diving species to permit higher concentrations of Mb to occur in tissues. How does this effect the data presented and should animals with this adaptation be analyzed separately?
Pg 3 line 52 change “very” to “vary”
Pg 5 line 100. What are “estimates” of body temperature, isn’t it an empirical measurement?
Pg 8, line 159. Delete “previously”
Pg 8 line 161. Change “analysis” to “analyses”

·

Basic reporting

I found the manuscript to be concise, well written, and easy to follow (with only a few minor spelling errors; see general comments). The introduction provides the relevant context and background rationale for the study. Figures and tables are suitable. All raw data is supplied as an appendix.

Experimental design

This article presents original research that lies within the scope of the journal. The background and research question is well defined, and builds upon previous (ambiguous) research in this area. Methods description is appropriate.

Validity of the findings

The data is robust (>200 species, >1000 individual values) and the analyses are, to the best of my ability to judge, statistically sound. The text is somewhat minimalist and solely focused on the key findings of the study (i.e. little to no speculation).

Additional comments

I enjoyed reading this concise, well written manuscript, and have no major concerns regarding the study design, data analyses, or conclusion. It is pleasing to see that ectotherms follow a similar allometric pattern to that repeatedly demonstrated for endothermic divers. A potential confounding factor for the ectothermic species, that was briefly mentioned (‘aquatic respiration’ – please replace with ‘cutaneous respiration’) by not addressed, is the degree of gas exchange by the integument – which should differ dramatically between amphibians and reptiles (though only one amphibian value was included). Also, I wonder if there are not dive duration values available for obligate air breathing fish (African lungfish, pirarucu, swamp eel) that could be included with the study?

Specific comments for improvement are provided below:

Line 52. Please replace “very” with “vary”.

Line 68. organisms --> organism

Line 138. Should “ranged from only 0.21 to 0.31” instead read ‘ranged from only 0.21 to 0.34” (as per table 2)?

Lines 153-155. Does the ‘1-2 orders of magnitude’ value quoted account for differences in body temperature (i.e. corrected to 37C)? Also, I assume the authors are referring to mass-specific (as opposed to total) oxygen consumption rates here? Please clarify.

Lines 158-159. Why is it surprising that dive duration varies with temperature? Is this not the expected result? I suggest rephrasing this sentence.

Appendix 1. It is not immediately clear what numbers in square parentheses refer to (reference numbers). Please include this in the appendix heading and, if possible, in the table itself.

·

Basic reporting

No Comments

Experimental design

No Comments

Validity of the findings

No Comments

Additional comments

Review: Hayward et al.

This paper develops a diving model based on metabolic rates and body temperature, which provides predictions across vertebrate groups of air-breathing divers. Consistent with the predictions they find that ectothermic divers, with lower metabolic rates, have longer dive times but also lower body-mass scaling of dive duration – possibly linked to a steeper metabolic scaling with respect to mass in this group.

Main points: overall, this is a clear and sensible paper which makes some novel (though intuitive) conclusions regarding the body-size scaling of dive duration.

1. Given the link between metabolic rate and dive duration, I expected to see a plot of the predicted relationship based on difference in metabolic rate between body mass and dive duration for ectotherm and endotherm divers of different mass. Can the model be used to make a specific predictions of the intercepts in the dive duration-mass relationships for the two metabolic groups? Could this be used against the data plotted in figure 1?
2. I find it odd that the authors accept 2/3 scaling of metabolic rate for endotherms. There has been so much debate about this in the past, and although 2/3 is consistent with the estimated scaling of dive duration of 1/3, it goes against the weight of evidence of a mode in scaling of ¾ seen in other major reviews – (e.g. Isaac, N. J. B., & Carbone, C. (2010). Why are metabolic scaling exponents so controversial? Quantifying variance and testing hypotheses. Ecology Letters, 13(6), 728–35). Might there be another factor to consider here? Might there be different selection pressures for diving vertebrates?

---

## Round 0.2 · accepted · Accept

Thanks for following through with the reviewers comments. I find your ms ready for publication. Congratulations!